# Bone Deformities through the Prism of the International Classification of Functioning, Disability and Health in Ambulant Children with Cerebral Palsy: A Systematic Review

**DOI:** 10.3390/children11020257

**Published:** 2024-02-16

**Authors:** Rodolphe Bailly, Christelle Pons, Anne-Charlotte Haes, Lisa Nguyen, Matthias Thepaut, Laëtitia Houx, Mathieu Lempereur, Sylvain Brochard

**Affiliations:** 1Pediatric Rehabilitation Department, Fondation Ildys, Rue Alain Colas, 29200 Brest, France; laetitia.houx@chu-brest.fr (L.H.); sylvain.brochard@chu-brest.fr (S.B.); 2Laboratoire de Traitement de L’Information Médicale (LaTIM), Inserm U1101, Université de Bretagne-Occidentale, 29200 Brest, France; christelle.ponsbecmeur@ildys.org (C.P.); annecharlotte.haes@gmail.com (A.-C.H.); lisa_nguyen@orange.fr (L.N.); matthias.thepaut@chu-brest.fr (M.T.);; 3Physical and Rehabilitation Medicine Department, University Hospital of Brest, 29200 Brest, France; 4Faculty of Medicine, Western Britany University, 29238 Brest, France; 5School of Physiotherapy (IFMK), CHRU Morvan, 29200 Brest, France; 6Pediatric Surgery Department, University Hospital of Brest, 29200 Brest, France

**Keywords:** bone deformity, ICF, gait, cerebral palsy, children

## Abstract

(1) Aim: The aim of this study was to determine the relationship between lower limb bone deformities and body functions, activity, and participation in ambulant children with CP and whether changing bone morphology affects outcomes in these domains. (2) Methods: A systematic literature search (PROSPERO CRD42020208416) of studies reporting correlations between measures of lower limb bone deformities and measures of body function, activity or participation, or post-surgical outcomes in these domains was conducted from 1990 to 2023 in Medline, Scopus, and Cochrane Library. We assessed study quality with the Checklist for Case Series (CCS) and a quality assessment developed by Quebec University Hospital. Meta-analysis was not possible; therefore, descriptive synthesis was performed. (3) Results: A total of 12 of 3373 screened articles were included. No studies evaluated the relationships between bone deformities and activity or participation, or the effect of isolated bone surgery on these domains. Correlations between bone deformities and body functions were poor-to-moderate. Internal hip rotation during gait improved after femoral derotation osteotomy. (4) Conclusions: A shift in paradigm is urgently required for the research and management of bone deformities in children with CP to include the activity and participation domains of the ICF, as well as consider more psychological aspects such as self-image.

## 1. Introduction

Cerebral palsy (CP) [1] affects all aspects of an individual’s life, as described by the International Classification of Functioning, Disability and Health (ICF) [2], the classification of health and health-related domains. The current models of rehabilitation for children with CP include restoring function but also extend to the goals of improving activity and participation [3]. To illustrate the ICF model, we can use the example of gait. The ability to walk (“motor function” in the ICF) facilitates integration into society (“participation” in the ICF), in particular through social interactions, and gait impairment is the strongest predictor of the level of activity and participation of children with CP [4]. Gait impairments are a common feature of CP [5] and are frequently clinically associated with lower limb bone deformities (“structures” in the ICF) [6,7,8,9]. A bone deformity is a structural deviation or distortion of a bone’s morphology from its normal alignment, length, and/or size (Deformity [Internet]. In: Venes DD, editors. Taber’s Medical Dictionary. F.A. Davis Company; 2021. Available online: https://www.tabers.com/tabersonline/view/Tabers-Dictionary/744438/4/deformity (accessed on 15 March 2022)). For example, in children with CP, femoral anteversion is reported to be between 25° and 45°, external tibial torsion between 20° and 30°, and the neck-shaft angle between 135° and 150° [6,7,8,9,10]. These values are commonly accepted as excessive and related to the child’s development [2], often described as induced by malposition (e.g., sitting in a W position), and the exact aetiologies remain poorly understood. Despite the fact that the range and prevalence of bone deformities in the lower limbs of children with CP are not well described, preventing the onset of bone deformities and treating existing bone deformities are major aims of rehabilitation and surgical interventions. The overall aim is to maintain efficient gait with minimal pain for as long as possible to support the child’s autonomy throughout his or her life [3,11]. Bone deformities are thus projected through the prism of ICF by therapists, in a “bottom-up approach” [12], to consider and prevent their impact on all aspects of the individual’s life.

The prevention of musculoskeletal disorders is a key component of care strategies (e.g., physiotherapy, orthoses, and botulinum toxin injections) [13] for children with CP from the earliest age, despite the lack of mid- and long-term evidence for the effectiveness of these interventions or care strategies on the occurrence of bone deformities. If bone deformities are not improved by conservative treatments, or if they continue to progress despite such interventions, surgery is typically proposed. Single-event multi-level surgery (SEMLS) is currently the most widely used procedure for the treatment of musculoskeletal disorders. It involves the correction of several bone and/or muscle disorders within the same surgical procedure, and it is usually performed with the aim of correcting gait deviations [3]. However, SEMLS requires long periods of postoperative rehabilitation, and it is associated with short- and medium-term postoperative pain, particularly when bone procedures are performed [14]. Therefore, the issue of whether or not to perform bone procedures is highly complex and is a major point of discussion between the surgeon and the rehabilitation professionals. The outcome of this decision has an impact on the whole rehabilitation process and has major implications on the life of the individual and their family in the short to mid-term (for instance: pain, immobilization, and long periods of postoperative rehabilitation).

To enlighten the decision-making process, it is essential to determine the precise relationship between bone deformity and body function, activities, and participation, as well as the consequences of altering bone morphology on these domains. This information would help clinicians to prioritize the different strategies to prevent bone deformities, to propose appropriate surgical bone interventions when necessary, and to manage gait rehabilitation. To shed light on these issues, we conducted a systematic review of the literature relating to bone deformities, surgical bone interventions, and the domains of the ICF. Our aim was to provide insights into the multifactorial relationships of the neuromusculoskeletal system through a focus on bone deformities. We chose to include only surgical studies that had evaluated single bone procedures, rather than SEMLS, to precisely report the relationships between changes in bone morphology and body functions, activity, and/or participation outcomes, without confounding factors related to associated procedures.

The aims of this systematic review were two-fold: (1) to report the evidence for relationships between lower limb bone deformities and body functions, activity, and participation in ambulant children with CP and (2) to report the effect of changes in bone morphology (induced by surgery) on these domains. From a clinical point of view, we hypothesized that bone deformities would be moderately correlated with outcomes from the body function, activity, and participation domains of the ICF, and that these correlations would be strengthened after a reduction in the bone deformities due to improvements in these domains.

## 2. Materials and Methods

We conducted a systematic literature search (type: overview, recorded on PROSPERO: CRD42020208416) according to the Preferred Reporting Items for Systematic Reviews and Meta-Analyses (PRISMA) guidelines [15] in order to answer two questions. Question 1: In ambulant children with CP, is there a relationship between lower limb bone deformities and body function, activity, and participation outcomes? Question 2: In ambulant children with CP, do changes in lower limb bone morphology (i.e., after surgery) positively impact on body function, activity, and participation outcomes?

### 2.1. Search Strategies and Resources

To answer these questions, we searched Medline, Scopus, and Cochrane Library databases from inception to September 2023. For question 1, we built a specific search string to identify studies that evaluated the relationship between lower limb bone morphology variables and body function (gait, strength, spasticity, etc.), activity (mobility, running, jumping, etc.), and participation (leisure activities, schooling, etc.) outcomes. Two reviewers (ACH and RB) conducted the database search independently, using the same search strings, to ensure completeness, and any disagreement was resolved by discussion between the 2 reviewers [16]. For question 2, we built a search string to identify studies that evaluated the pre–post effect of a single lower limb bone surgical procedure on outcomes relating to body functions, activity, and participation. Two reviewers (LN and RB) conducted this database search independently, using the same search strings, to ensure completeness, and any disagreement was resolved by discussion between the two reviewers [16]. Full details of the terms used in both searches are provided in Appendix A. A backward citation search was also performed by checking the references of the articles included.

For each question, the 2 reviewers screened all titles and abstracts retrieved for eligibility. If the information in the abstract was not sufficiently clear to determine eligibility, the full text was downloaded. Agreement (Kappa) between the 2 reviewers was verified at these stages.

### 2.2. Inclusion and Exclusion Criteria

Question 1: We included all studies that met the following criteria:-Included ambulant children aged 0 to 18 years with any type of CP;-Used an objective measurement (clinical examination or imaging) of at least 1 lower limb bone variable (for instance, neck-shaft angle, tibial torsion, etc.);-Used a standardized assessment of 1 or more body function (i.e., spasticity, muscle strength, etc.), activity (i.e., moving around in different locations, dressing, etc.), and participation (i.e., shopping, etc.) outcomes;-Used a statistical analysis of correlations (i.e., statistical report of the relationship with R, R^2^, etc.) between bone morphology variables and body function, activity, and participation outcomes.

Question 2: We included all studies that met the following criteria:-Included ambulant children aged 0 to 18 years with any type of CP who underwent a single bone surgical procedure;-Evaluated a single bone surgery (excluding SEMLS);-Had a pre–post design to assess change after surgery;-Involved a statistical evaluation of the impact of the procedure on at least 1 body function, activity, and participation outcome (measured using a standardized assessment).

Studies had to be published in a peer-reviewed journal; all types of studies could be included to provide a broader overview of the state of evidence on both questions; and studies that were unavailable, including after contacting the authors, were excluded.

### 2.3. Quality Evaluation

For the first question, the studies included all fulfilled the criteria for consideration as a case series (study in which only patients with the disease are sampled [17]); thus, we used the JBI critical appraisal Checklist for Case Series (CCS) [18] to determine their quality. The CCS assesses the risk of bias, the quality of the data collected, and the relevance of the statistical analysis. To best evaluate the included studies, we divided the outcome item into 4 different items (2 related to bone deformities and 2 to body functions, activity, and participation; for both cases we report the assessment used and its validity). We also divided the statistical item into 2 items (rationale for the choice of test and criterion for analysis of correlation results). The addition of items to the CCS is authorized by the authors of the checklist. For the second question, we used a quality assessment grid for observational studies developed by Quebec University Hospital [19] to assess the quality of the information reported on the characteristics of the bone surgery protocols and rehabilitation follow-up (description, compliance, etc.). To make the grid more relevant to our research question, and as authorized by the authors, we added one item (M9: Is the postoperative management protocol described?), modified item R7 (to refer to comparison with the control group), and removed questions R3, R5, A3, and S1, which were not relevant to this review. Each reviewer completed the forms independently and agreement was verified (Kappa). Any disagreement was resolved by discussion. The total quality score obtained after agreement was normalized to 100 to produce a Q-score from 0 to 100 for each study.

### 2.4. Data Extraction

In accordance with the Cochrane recommendations, relevant data were independently reported on data extraction sheets by ACH and RB for question 1 and by LN and RB for question 2 [20]. Data extracted from studies included the authors, date of the study and institution; demographic and clinical data; the purpose and the type of study; the surgical procedure (when relevant) and postoperative follow-up (pain and immobilization) when reported; the variables and outcomes assessed and the method of assessment; and the statistical tests and the strength of the correlations or the results of statistical comparisons.

### 2.5. Analysis

The large heterogeneity of the evaluation methods and surgical procedures prevented the pooling of data for a meta-analysis. We therefore performed a descriptive synthesis based on the correlations reported and the quality of the studies included. The strength of correlations was defined according to Altman’s criteria [21] (R < 0.20: poor; 0.21–0.40: fair; 0.41–0.60: moderate; 0.61–0.80: good; 0.81–1.00: very good). The sign of the correlations was adjusted according to the direction of the measurements to respect the following convention: positive sign for external rotations and negative for internal rotations.

## 3. Results

### 3.1. Study Selection

Question 1: A total of 1711 titles and abstracts were identified, of which 10 studies [10,22,23,24,25,26,27,28,29,30] (*n* = 773 individuals; mean age: 11 years; topography: *n* = 180 unilateral CP (UCP), *n* = 430 bilateral CP (BCP), and *n* = 51 unknown (UNK); Table 1) met the inclusion criteria (Figure 1). We note that two additional studies [31,32] used principal component analysis (PCA) and reported indirect relationships between lower limb bone deformities and gait through predictive models. It was not possible to extract direct relationships between one bone morphology variable and the domains of the ICF; therefore, these studies were excluded. Quality ratings of the 10 included studies are shown in Appendix A: Q-scores ranged from 36 to 75 out of 100 (Appendix A). Interrater agreement was strong: selection by title K = 0.92 and selection by abstract K = 0.9.

Question 2: Out of 1662 titles and abstracts, 2 studies met the inclusion criteria [33,34] (Figure 1). They included *n* = 152 individuals with a mean age of 10 years, all with bilateral CP (Table 1). Quality ratings are shown in Appendix A: quality ratings were 59 and 60 out of 100, respectively (Appendix A). Interrater agreement was strong: selection by title K = 0.91 and selection by abstract K = 0.87.

**Table 1 children-11-00257-t001:** Description of the studies included.

Author	Year	*n*	Age	CP Type	*n*	GMFCS (*n*)	Sex *n* (%)	Bone Morphology Variable	Assessment
Teixeira et al. [30]	2018	195	10.2 (3–18)	Unilateral	43	I (61) II (90) III (44)	Female 86 (44)	TT	PE (TA)
				Bilateral	152		Male 109 (56)		
Westberry et al. [24]	2018	77	11.8 (7.2–18.7)	Unilateral	30	I–II	Female 28 (36)	FT	EOS
				Bilateral	47		Male 49 (64)		
Cho et al. [22]	2018	57	At Physical Exam: 3.6 ± 1.6 (2–6)	Unilateral	10	I (20) II (13) III (10) IV (11) V (3)	Female 26 (46)	NSA	3D CT
			At imaging study: 9.2 ± 1.8 (7–14)	Bilateral	47		Male 31 (54)	FT	
Presedo et al. [29]	2017	114	12.1 ± 0.3 (5.5–19.2)	Bilateral	114	I (6) II (67) III (41)	Female 47 (41)	FT	PE (Ruwe)
							Male 67 (59)	TT	PE (TA)
Kim et al. [27]	2017	26	12.6 (6–16)	Bilateral	26	-	Female 12 (46)	FT	3D CT
							Male 14 (54)	TT	3D CT, PE (TFA)
Karabicak et al. [23]	2016	20	12.3 ± 4.5	Unilateral	9	I (1) II (6) III (4) UK (9)	Female 8 (40)	FT	PE (Ruwe)
				Diplegia	6				
				Triplegia	1		Male 12 (60)		
				Quadriplegia	4				
Lee et al. [10]	2013	33	9.5 ± 6.9	Bilateral	33	I (15) II (18)	Female 13 (39)	FT	3D CT
							Male 20 (61)	TT	
Desloovere et al. [26]	2006	200	8.1 ± 2.4	Unilateral	88	-	-	FT	PE (Ruwe)
				Bilateral	112	-	-		
Kerr et al. [25]	2003	29	14.6 (4.6–35.8)	-	-	-	Female 11 (38)	FT	PE (Ruwe)
				-	-	-	Male 18 (62)		
Aktas et al. [28]	2000	22	13.7 (6.4–20.6)	-	-	-	Female 6 (27)	FT	3D CT
				-	-	-	Male 16 (73)	TT	3DCT, PE (TA, TFA)
Boyer et al. [34]	2017	140	9.4 ± 4.0 (3.7–17.2)	Bilateral	140	I (52) II (55) III (4)	Female 63 (45)	Femoral derotation osteotomy	
							Male 77 (55)		
Cimolin et al. [33]	2011	12	11.7 ± 3.4	Bilateral	12	-	Female 6 (50)	Femoral derotation osteotomy	
							Male 6 (50)		

GMFCS: gross motor function measure; TT: tibial torsion; FT: femoral torsion; NSA: neck-shaft angle; PE: physical examination; TA: transmalleolar axis method; CT: computed tomography, TFA: thigh foot angle.

The reasons for exclusion after full reading are reported in Appendix A (question 1) and Appendix A (question 2).

The outcomes were classified according to the ICF [35] (Table 2); thus, the evaluation of gait in a movement analysis laboratory was graded as “b770” and assigned to the item “Neuromusculoskeletal and movement-related functions” of the “Body Functions” component.

### 3.2. Bone Morphology Variables Evaluated in the Studies Included

Among the different deformities reported for the individuals with CP, only three bone morphology variables were evaluated in the studies related to question 1: neck-shaft angle (NSA) (one study [22]; *n* = 57; Q-score = 54), femoral torsion (FT) (nine studies [10,22,23,24,25,26,27,28,29]; *n* = 578; Q-score = 36–75), and tibial torsion (TT) (five studies [10,27,28,29,30]; *n* = 390; Q-score = 36–68). For question 2, both studies (*n* = 152; Q-score = 59–60) evaluated the effect of a femoral derotation osteotomy (FDO) on gait outcomes assessed by three-dimensional gait analysis (3DGA) [33,34]. Cimolin et al. [33] explored the effect at 10 months post-op (*n* = 12; Q-score = 59), while Boyer et al. [34] evaluated the short- (9–24 months post-op, *n* = 140; Q-score = 60) and mid-term effects (>36 months post-op, *n* = 29). These bone variables were assessed using different methods depending on the study’s objective: physical examination (seven studies [23,25,26,27,28,29,30]; *n* = 606; Q-score = 36–71), radiographic measurements (one study [22]; *n* = 57; Q-score = 54), three-dimensional computed tomography (3D CT) (three studies [10,27,28]; *n* = 81; Q-score = 36–68), and biplanar radiography (EOS^®^) (one study [24]; *n* = 77; Q-score = 75), (Table 2).

### 3.3. Relationships between Bone Deformity and ICF

The following section presents the findings that answer question 1: “In ambulant children with CP, is there a relationship between lower limb bone deformity and body function, activity and participation outcomes?”

#### 3.3.1. Relationships between Lower Limb Bone Deformity and Body Function Outcomes

Ten studies evaluated the relationship between at least one lower limb bone morphology variable and one body function outcome. An overview of the results can be found in Figure 2 and Table 2.

Neck-Shaft Angle (NSA)

One study [22] (*n* = 57; Q-score = 54) examined the relationship between NSA and a body function outcome, in this case, spasticity. NSA was assessed by radiological measurement. The relationship between NSA and spasticity was overall poor-to-moderate (0.18 to 0.56 depending on the muscle) with a moderate correlation between NSA and adductor spasticity.

Femoral Torsion (FT)

Nine studies [10,22,23,24,25,26,27,28,29] (*n* = 578; Q-score = 36–75) evaluated the relationship between FT and a body function outcome; gait outcomes were most commonly evaluated in the body function domain. FT was measured by physical examination [23,25,26,29], EOS [24], or 3D CT [10,22,27,28]. Correlations between FT and body functions were poor-to-moderate.

Seven studies [10,24,25,26,27,28,29] (*n* = 501; Q-score = 36–75) evaluated the relationship between FT and at least one gait outcome assessed by 3DGA. FT and kinematic outcomes were poorly to moderately correlated (R = 0.01 to 0.48). For instance, the relationship between FT and hip rotation was poor-to-moderate (R = 0.01 to 0.46) (*n* = 387) [10,24,25,26,27,28]. One study [26] (*n* = 200; Q-score = 57) evaluated the relationship between FT and spatiotemporal gait outcomes (cadence, velocity, and step length) and reported no significant correlations. One study evaluated the relationships between FT and kinetic outcomes. They found only one significant correlation: between FT and the timing of the zero-hip moment. One study [22] (*n* = 57; Q-score = 54) reported no correlation between FT and hamstring and adductor muscle spasticity. One study [23] (*n* = 20; Q-score = 71) described a poor-to-moderate relationship between FT and balance. The strongest relationship found for FT (R = 0.46) was with dynamic reaching assessed by the Trunk Control Measurement Scale.

Tibial Torsion (TT)

Five studies [10,27,28,29,30] (*n* = 390; Q-score = 36–68) evaluated the relationship between TT and at least one gait outcome assessed by 3DGA. In these studies, TT was measured by physical examination (various methods: transmalleolar axis [28,29,30] (*n* = 331; Q-score = 36–61), thigh foot angle [27,28] (*n* = 48; Q-score = 36), and by 3D CT [10,27,28] (*n* = 81; Q-score = 36–68)).

Poor-to-moderate correlations were reported between TT and body function outcomes (R = 0.15 to 0.72; Q-score = 36–68). Overall, the relationship between TT and foot progression angle was the most evaluated (three studies [10,29,30], *n* = 342; Q-score = 61–68). Correlations between TT and foot progression angle were poor-to-moderate (R = 0.15 to 0.56; Q-score = 61–68) [10,29,30]. Correlations were moderate between TT and tibial torsion during gait (R = 0.61 to 0.70; Q-score = 36) [28] and between TT and knee rotation during gait (R = 0.62 to 0.72; Q-score = 36) [27]. Correlations between TT and pelvic, hip, and knee rotation during gait were poor (R = 0.06 to 0.22; Q-score = 68) [10].

#### 3.3.2. Relationships between Deformity and Activity or Participation Outcomes

No studies evaluated the relationship between a lower limb bone morphology variable and an activity or participation outcome.

### 3.4. Results following Isolated Bone Surgery

The following section presents the findings that answer question 2: “In ambulant children with CP, do changes in lower limb bone morphology (i.e., after surgery) impact on body function, activity and participation outcomes?”

#### Body Function

An overview of the results can be found in Figure 3 and Table 3.

**Figure d66e1325:**
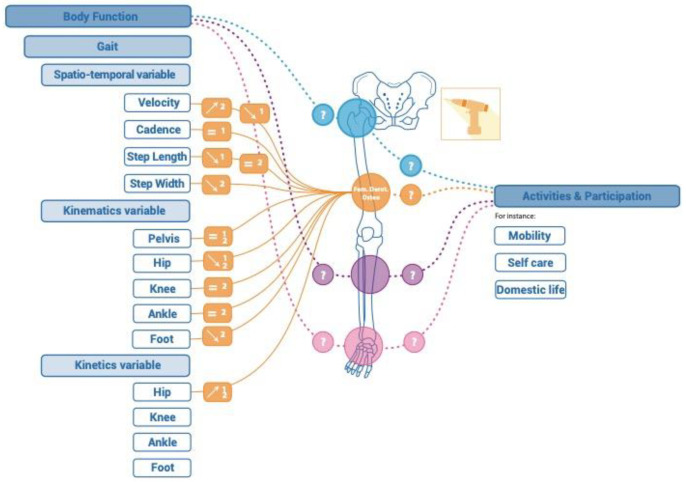


The effects of FDO on spatiotemporal variables differed between the two studies for gait velocity and step length in the short and mid-term (*n* = 152; Q-score = 59–60). Boyer et al. [34] showed a decrease in gait velocity associated with a decrease in step length in the short and mid-term, whereas Cimolin et al. found an increase in gait velocity associated with a trend of an increase in step length in the short term. At the same time, Cimolin et al. [33] showed a decrease in step width (*n* = 12), whereas Boyer et al. [34] found no effect on cadence in the short (*n* = 140) and mid-term (*n* = 29).

Both studies reported a lack of effect of FDO on pelvic, knee, and ankle motion during gait. However, both reported a significant decrease in internal hip rotation during gait in the short (*n* = 41) and mid-term (*n* = 29). Cimolin et al. [33] found a significant decrease in the foot mean progression angle in the short term (*n* = 12).

Boyer et al. [34] found no significant change in the hip abductor moment in the short term (9 to 24 months post-op, *n* = 140) and a significant increase at mid-term (>36 months post-op, *n* = 29). Cimolin et al. [33] (*n* = 12) found a significant increase in the hip extensor moment arm in the short term (10 months post-op).

##### 3.4.2. Activity and Participation

No studies evaluated the effect of a single lower limb bone surgical procedure on activity or participation in ambulant children with CP.

## 4. Discussion

In this review we used the ICF framework to determine the impact of lower limb bone deformity on the lives of ambulant children with CP. We also sought out studies that reported the effect of changes in single anatomical bone characteristics on these domains. Regarding our aim to identify data describing the relationships between bone deformity and the ICF outcomes, we were unable to verify our hypothesis that bone deformity would be moderately correlated with outcomes from the activity and participation domains of the ICF since we found no studies that evaluated any aspects of activity or participation. However, in contrast with our hypothesis, we found evidence of poor-to-moderate correlations between anatomical lower limb bone characteristics and the body functions domain. It is interesting to note that only three anatomical bone characteristics (NSA, FT, and TT) have been studied in the literature. This may be because these variables are considered particularly important to monitor bone deformities and to determine treatment strategies and also because surgical procedures to modify them exist [2]. The strongest relationships found for these variables, in the studies with the best Q-scores, were between femoral torsion (FT) and hip rotation during gait (R = 0.46) and between tibial torsion (TT) and foot progression angle during gait (R = −0.46 to 0.56).

Regarding our aim to identify data describing the impact of isolated bone surgery on the ICF outcomes, we did not find any studies that evaluated the impact on activity and participation. Two studies evaluated the impact on body functions: they showed a significant decrease in internal hip rotation during gait after FDO. These limited results suggest that there is little association between bone deformities and gait outcomes.

The gaps in our search results clearly highlight that there has been a considerable lack of interest in the impact of bone deformity, and its management, on the activity and participation of children with CP, despite the fact that activity and participation are the ultimate goals of prevention and cure strategies.

### 4.1. Bone Morphology and Body Functions

From a specific point of view, the analysis of the literature shows a moderate correlation between dynamic hip rotation during gait and the value of FT. When the rehabilitation objective is specifically to modify dynamic hip rotation during gait, it may be relevant to evaluate the opportunity of femoral derotation surgery and to discuss it with the child and the family. The literature reports that this specific bone surgery makes it possible to correct the angle of hip rotation during the gait cycle. This way of thinking could, for example, also be applied to the moderate relationship between the angle of foot progression and TT, leading to a proposal for bone surgery with the specific aim of modifying this angle.

From a more global point of view, the accepted wisdom that there is a strong relationship between bone deformity and gait deviations in children with CP [2,36] is not supported by the results of this review. In fact, the results showed evidence to the contrary: the relationships between both FT and dynamic hip rotation during gait [10,24,25,26,27,28] and between TT and foot progression angle [10,29,30] are, at best, poor-to-moderate. Furthermore, evidence demonstrated a lack of a relationship between FT and spatiotemporal gait outcomes [26]. This is supported by two other studies in this domain that also found few relationships between bone deformity and gait deviations using PCA [31,32]. Taken together, therefore, the existing literature suggests that gait deviations are multifactorial, impacted by other primary and secondary symptoms of CP such as spasticity, muscle weakness, and poor trunk control [37,38,39]. But there is little evidence to support any link between these symptoms and lower limb bone deformity (i.e., between FT and spasticity [22] or trunk control [23]). These results also indicate that these biomechanical interactions are specific to each individual. An individual assessment combining the monitoring of spatiotemporal, kinematic, and kinetic gait parameters using 3D gait analysis [5,40] with a standardized assessment of bone deformities (CT scan or EOS) would increase the understanding of individual biomechanical interactions and the effects of interventions affecting those interactions. However, this relationship between gait deviations and deformities cannot be generalized to all children with CP. Future research should focus on specific profiles of ambulant children for whom bone deformities are expected to have a greater impact on gait, for example, children with GMFCS level III or even more specific groups.

Surgical goals are often focused on normalizing the gait pattern or increasing range of motion [41,42], and most SEMLSs involve a similar proportion of bone and muscle procedures. The postoperative consequences of a bone intervention are considerable: procedures are often associated with significant secondary effects for the child, including short-term postoperative pain, immobilization, and long periods of postoperative rehabilitation. According to the results of our review, FDO as a single surgical procedure has mixed outcomes in terms of spatiotemporal, kinematic, and kinetic gait outcomes. Despite the widespread use of this procedure and the uncertainty of its effects at a group level, research to refine the indications for the procedure is currently non-existent. We found no study that evaluated the effects of single surgical bone procedures on the activity or participation of children with CP. The level of interest in the effect of SEMLS and participation is also very low. A recent review of 74 (*n* = 3551) studies of SEMLS found only 3 studies (*n* = 125) that evaluated its effect on participation [43]. We conclude, therefore, that despite its importance for the child and their family, increasing activity and participation is not yet an area of interest for musculoskeletal surgery research.

### 4.2. Bone Deformities and Activity/Participation Outcomes

Clinical practice uses a bottom-up approach, and clinicians frequently seek to identify the relationships between lower limb bone deformities and gait deviations to determine treatment strategies that will improve gait at the body function level [33]. In contrast, the results of this review showed that the direct impact of bone deformities and/or its management on activity and participation has been only occasionally considered. For example, a child with CP, excessive femoral torsion, and an in-toeing gait (body functions) may be unable to run (activities) and play soccer with his or her friends (participation). The question then is the following: “What is really stopping this child from playing football with their friends, when they really want to?” It is currently impossible to determine whether the limiting factor is the excessive femoral torsion, so the question remains unresolved, although the solution often suggested is surgery. These relationships need to be studied in more detail to provide an evidence base for the preventive and curative treatment of bone deformities.

### 4.3. Towards a Paradigm Shift in Bone Deformity Prevention and Interventions for Ambulant Children with CP

The results of this review highlight the urgent need for a paradigm shift in the research and rehabilitation focus in this area. In the light of current models of care supported by the WHO, namely the ICF, it is no longer acceptable to focus purely on “structures” and “body functions”. The impact of all treatments, particularly long-lasting prevention or invasive and permanent interventions such as bone surgery, should be evaluated on all the domains of the ICF. In our opinion, research must focus on three main areas of equal importance to support the development and implementation of appropriate interventions to help the child to achieve his/her goals. First, an important goal of medical and surgical interventions in ambulant children with CP must be to improve social participation, as decided by the child and their family. Improving activity and participation are considered as essential outcomes of lower limb orthopaedic surgery by children with CP and their parents [44]; however, surgeons are currently unsure of the optimal methods to measure outcome following lower limb orthopaedic surgery in children [45]. Although this may seem a utopic goal, it seems necessary to start by reducing discomfort and stabilising or improving gait patterns, which in combination with other factors (e.g., improved self-esteem) could lead to better social participation for people with CP. Secondly, research is necessary to understand the extent to which the normalization of bone deformities prevents pain [46] and the development of osteoarthritis [47]. Adults with CP have more osteoarthritis than adults without CP [47]; however, the relationship between bone deformity and the development of pain and/or osteoarthritis has not yet been clearly established. A third important issue concerns self-image. It is already known that adolescents with CP have a less favourable view of their body, compared to able-bodied peers [48,49]. Furthermore, CP is known to cause particular difficulties in forming intimate and sexual relationships [50]. The final aim desired by the individual in improving their gait pattern or speed, reducing pain, or improving self-image may well be to improve social participation. To reach this goal, all these issues must be explored and considered in the management of each individual child, including decisions to undertake permanent bone-modifying procedures.

### 4.4. Limitations

Although SEMLS is the most commonly performed intervention for children with CP, we chose to focus this review on studies of single bone surgery interventions because the large heterogeneity of the profiles of the patient groups and different surgical procedures performed in SEMLS would make it impossible to determine the effects of any one specific bone intervention. Specifically, the aim of this review was to precisely determine current knowledge of the relationships between changes in bone morphology and body functions, activity, and participation. These specific relationships are currently only reported in studies of surgical interventions, and they can only be interpreted if soft tissue procedures have not been concomitantly performed. This question was designed to complement question 1 of this review, which examined knowledge of the relationships between bone deformities and these domains at a single time-point.

The reader should be aware that the profiles of children who undergo isolated bone surgery might be significantly different from those who undergo SEMLS. It is also important to highlight that the correlations reported cannot not be interpreted as relationships of causality of one variable on another. The results of this review should therefore be interpreted with caution. A large, well-designed study should be conducted to clarify the question of the causality of primary disorders on the development of secondary and tertiary disorders in patients with CP.

Although this review has provided an overview of the relationships between bone deformities and body function, activity, and participation, the large heterogeneity of the data included in terms of samples, assessments, and methodologies prevented meta-analysis and a more detailed interpretation. Moreover, the heterogeneity of the quality of the included studies limits generalization of the results. The results of this study are applicable to ambulant children with CP (mostly children with unilateral CP with a GMFCS level of I to III) and thus are not generalizable to non-ambulant children for whom the management of musculoskeletal disorders must be specifically considered. Readers should also be aware that these relationships may be different depending on the child’s GMFCS level. Unfortunately, it is impossible to perform a subgroup extraction of the data present in the literature. The small number of articles selected, in relation to the number of occurrences, could appear to be a limitation. This small number is partly due to the selection criterion, which required a specific statistical evaluation of the relationship between the bone variables evaluated and at least one ICF outcome. Nevertheless, this criterion was necessarily strict to enable the questions posed to be answered. The absence of articles evaluating the relationship between bone deformities and activity/participation in children with cerebral palsy may also explain the low number of articles selected, but in itself it constitutes a message that opens the way to new studies evaluating these relationships. A better understanding of bone deformities (in terms of prevalence and the link with the topography of the disorder, for example) and of their relationship with activity and participation would guide the choice of appropriate interventions for children with CP.

Although we did not exclude any papers because of language, the fact that the search string was based on the English language may have prevented some articles from being found.

## 5. Conclusions

The results of this review show that a shift in paradigm is urgently required for research into the impact of bone deformity and changes in anatomical bone characteristics through surgery on children with CP to include the activity and participation domains of the ICF, as well as to consider more psychological aspects such as self-image. In view of the weak relationship found between lower limb bone deformities and gait outcomes, we suggest that the orthopaedic status of ambulant children with CP should be regularly monitored. Given the few pieces of evidence regarding the links between structural deficits, function, activity, and participation, the assessment and follow-up of any ambulant child with cerebral palsy should include a precise assessment of the needs of the child and their family in terms of activity limitation and participation restriction. If this reveals that problems relating to gait are highlighted, or if the clinicians identify any orthopaedic or gait-related problems, the following should be regularly monitored:-Gait parameters (standardized assessment using 3D gait analysis which accurately monitors changes in spatiotemporal, kinematic, and kinetic gait parameters);-Orthopaedic parameters (e.g., bone deviations, spasticity, muscle strength, and length).

Surgical bone interventions should be limited to individual cases in which a biomechanical relationship has been clearly established between bone deformities, gait capacity, gait quality, and the desired level of activity and participation, including self-esteem.

This systematic review paves the way for future research, highlighting the need for large-scale, high-quality studies to assess the long-term effects of bone deformities and surgical interventions on the functional outcomes, activities, and participation of children with CP. Qualitative research to explore the psychosocial aspects of bone deformities in children with CP appears necessary to guide the choice of appropriate interventions for these children.

## Figures and Tables

**Figure 1 children-11-00257-f001:**
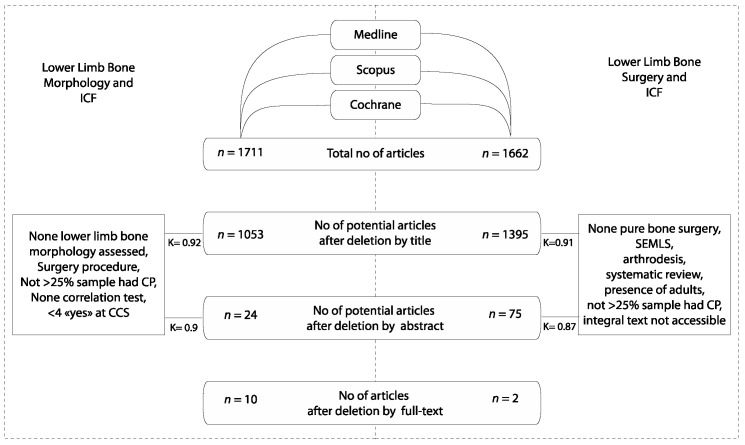
Row diagram of the included articles. K = kappa coefficient of inter-rater reliability.

**Figure 2 children-11-00257-f002:**
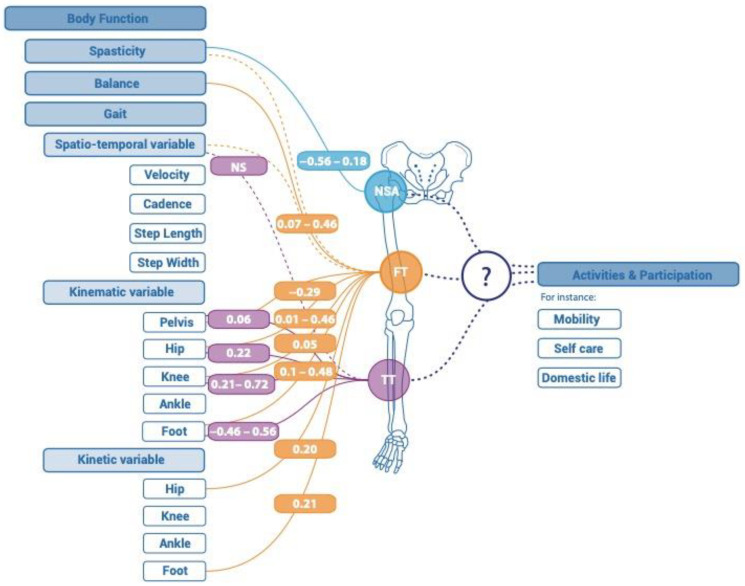
Correlations reported between lower limb bone morphology variable and body function, activity and participation for ambulant children with cerebral palsy. NSA: neck shaft angle; FT: femoral torsion; TT: tibial torsion, ＿ Significant relationship; ﹍ insignificant relationship; … no study.

**Figure 3 children-11-00257-f003:**
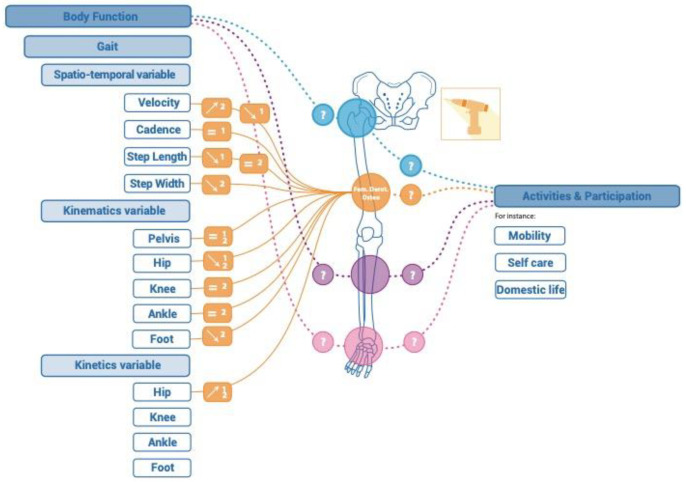
results reported after isolated bone surgery of the lower limbs of ambulant children with cerebral palsy. In blue: hip region; in orange: femoral region; in velvet: tibial region; in pink: ankle region; ＿ Presence of results in the literature; ﹍ Absence of results in the literature; ⬈ Isolated bone surgery increases the variable; ⬊ Isolated bone surgery decreases the variable.

**Table 2 children-11-00257-t002:** Correlations between lower limb bone morphology variables and body function, activity, and participation outcomes in ambulant children with cerebral palsy.

Authors	Year	Body Function			Correlation		Bone Morphology Variable		
		Denomination	ICF Code	Assessment			Denomination	ICF Code	Assessment
Cho et al. [22]	2018	Age at imaging study			−0.33		NSA	s750	Radiographic measurement
		Spasticity of hamstring muscles: R1 (Muscle reaction)	b735	Modified Tardieu Scale	0.25				
		Spasticity of hamstring muscles: R2 (Full PROM)			0.18				
		Spasticity of adductor muscles: R1 with knee extension			−0.45				
		Spasticity of adductor muscles: R2 with knee extension			−0.56				
		Spasticity of adductor muscles: R1 with knee flexion			−0.47				
		Spasticity of adductor muscles: R2 with knee flexion			−0.36				
Cho et al. [22]	2018	Spasticity of hamstring muscles: R1 (Muscle reaction)	b735	Modified Tardieu Scale	−0.20		Femoral Torsion	s750	3D CT
		Spasticity of hamstring muscles: R2 (Full PROM)			0.07				
		Spasticity of adductor muscles: R1 with knee extension			0.14				
		Spasticity of adductor muscles: R2 with knee extension			0.17				
		Spasticity of adductor muscles: R1 with knee flexion			0.07				
		Spasticity of adductor muscles: R2 with knee flexion			0.16				
Karabicak et al. [23]	2016	TCMS—Total	b755	Functional Balance Evaluation	0.28				Physical Exam.
		TCMS—Static sitting balance			0.07				
		TCMS -Selective movement control			0.26				
		TCMS—Dynamic Reaching (dynamic trunk control)			0.46				
		PBS			0.25				
Westberry et al. [24]	2018	Internal Hip Rotation	b710	Clinical examination	0.25				EOS
		External Hip Rotation	b710		−0.30				
		Hip Rotation Static Motion	b755	3DGA	0.12				EOS
		Hip Rotation Dynamic Motion	b770		0.07				
Kerr et al. [25]	2003	Max internal hip rotation throughout the gait cycle (maxIR)		3DGA	0.43				Physical Exam.
		Max internal hip rotation in the stance phase (maxIRst)			0.47				
		Mean hip rotation in gait (mean)			0.44				
		Mean hip rotation in stance (meanst)			0.46				
		Minimum internal (or maximum external) hip rotation in gait (minIR)			0.46				
		Minimum internal (or maximum external) hip rotation in stance (minIRst)			0.46				
Desloovere et al. [26]	2006	Hip rotation angle at IC		3DGA	0.28				Physical Exam.
		Hip rotation angle at TO			0.29				
Kim et al. [27]	2017	Hip Rotation		3DGA	0.30				3D CT
Aktas et al. [28]	2000	Hip Rotation		3DGA	0.01				3D CT
Lee et al. [10]	2013	Hip Rotation		3DGA	0.38				3DCT
		Pelvic Rotation			−0.29				
		Knee Rotation			0.05				
		Foot Progression Angle			0.22				
		Adjusted Foot Progression Angle			0.35				
Presedo et al. [29]	2017	Foot Progr: Internal Group (*n* = 140 limbs)		3DGA	0.10				Physical Exam.
		Foot Progr: Internal Group, Plantar Contact (*n* = 60 limbs)			0.04				
		Foot Progr: Internal Group, Forefoot Contact (*n* = 80 limbs)			0.18				
		Foot Progr: External Group (*n* = 33 limbs)			0.30				
		Foot Progr: External Group, Plantar Contact (*n* = 15 limbs)			0.48				
		Foot Progr: External Group, Forefoot Contact (*n* = 18 limbs)			0.32				
Desloovere et al. [26]	2006	Timing of Toe Off		3DGA	0.21				Physical Exam.
		Foot mean alignment ST			0.29				
		Hip timing of 0 moment			0.20				
		Cadence			NS				
		Velocity			NS				
		Step Length			NS				
		Kinetics parameters (Hip timing at 0 moment)			NS to 0.2				
Lee et al. [10]	2013	Pelvic Rotation	b770	3DGA	0.06		Tibial Torsion	s750	3D CT
		Hip Rotation			0.22				
		Knee Rotation			−0.21				
Kim et al. [27]	2017	Knee Rotation		3DGA	0.62				3D CT
		Knee Rotation			0.72				Physical Exam. Thigh Foot Angle
Aktas et al. [28]	2000	Tibial rotation in Gait		3DGA	0.70				3D CT
		Tibial rotation in Gait			0.65				Physical Exam. Transmalleolar Axis
		Tibial rotation in Gait			0.61				Physical Exam. Thigh Foot Angle
Teixeira et al. [30]	2018	Foot progression at IC		3DGA	0.44				Physical Exam. Transmalleolar Axis Left Side
		Mean foot progression in St			0.49				
		Mean foot progression in single support			0.5				
		Max foot progression (Int Rot)			0.46				
		Min foot progression (Ext rot)			0.51				
		Mean foot progression in swing			0.48				
		Foot progression at IC		3DGA	0.49				Physical Exam. Transmalleolar Axis Right Side
		Mean foot progression in St			0.54				
		Mean foot progression in single support			0.54				
		Max foot progression (Int Rot)			0.52				
		Min foot progression (Ext rot)			0.56				
		Mean foot progression in swing			0.54				
Presedo et al. [29]	2017	Foot Progr: Internal Group (*n* = 140 limbs)		3DGA	−0.24				3 D CT
		Foot Progr: Internal Group, Plantar Contact (*n* = 60 limbs)			−0.33				
		Foot Progr: Internal Group, Forefoot Contact (*n* = 80 limbs)			−0.29				
		Foot Progr: External Group (*n* = 33 limbs)			−0.27				
		Foot Progr: External Group, Plantar Contact (*n* = 15 limbs)			−0.46				
		Foot Progr: External Group, Forefoot Contact (*n* = 18 limbs)			−0.15				
Lee et al. [10]	2013	Foot Progression Angle		3DGA	−0.34				3D CT
		Adjusted Foot Progression Angle			−0.33				
Absence of study		**Activity**			Unknown				
Absence of study		**Participation**			Unknown				

3DCT: three-dimensional computed tomography, 3DGA: three-dimensional gait analysis, EOS: biplanar radiography, Physical Exam.: physical examination. Strength of the correlation: 

0.0–0.2; 
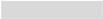
0.21–0.4; 
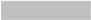
0.41–0.6; 

0.61–0.8; 
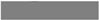
0.81–1.0. * corresponded to correlation with *p* > 0.01. As the information was missing for several studies and the *p*-value added confusion to the interpretation of correlations, the authors suggest deleting this irrelevant information.

**Table 3 children-11-00257-t003:** Results reported after isolated bone surgery of the lower limbs of ambulant children with cerebral palsy.

**Surgical Intervention**				**Spatiotemporal Variable**				**Kinematic Variable**								**Bone Measurement**			
								**Trunk**	**Pelvis**			**Hip**							
**Surgical Procedure**	**Study**	**Q-Score** **/100**	** *n* **	**Group**	**Velocity**	**Step Length**	**Step Width**	**Cadence**	**Trunk.Mean.Obl**	**Pelv.Mean.Obl**	**Pelv.Mean.Tilt**	**Pelv.Mean.Rot**	**Hip.Flex.Ext.IC**	**Hip.Min.Flex.St**	**Hip.Mean.Rot.Int.Ext**	**Hip.Add.Abd**	**Values**	**Assessment Method**	**Bone Variable**
Femoral Derotation Osteotomy	Boyer et al., 2017 [34]	60	*n* = 140	Preoperative	0.38 ± 0.09	0.71 ± 0.14	-	0.55 ± 0.08	−4.4 ± 5.5	1.4 ± 5.2	-	-	-	-	10.9 ± 13.1	-	50 ± 15	PE (Ruwe)	FT
			*n* = 140	Short Term (9–24 m post-op)	0.37 ± 0.08 *	0.65 ± 0.14 *	-	0.55 ± 0.08	−5.7 ± 7.5	0.8 ± 4.8	-	-	-	-	−2.0 ± 12.5	-	15 ± 10 *		
			*n* = 29	Preoperative	0.39 ± 0.10	0.72 ± 0.14	-	0.55 ± 0.12	−4.6 ± 2.5	1.9 ± 7.1	-	-	-	-	11.3 ± 12.8	-	50 ± 15		
			*n* = 29	Short Term (9–24 m post-op)	0.37 ± 0.11	0.62 ± 0.08	-	0.58 ± 0.12	−6.0 ± 6.2	1.1 ± 5.2	-	-	-	-	−2.7 ± 11.7 *	-	15 ± 10 *		
			*n* = 29	Mid-Term (>36 m post op)	0.34 ± 0.06 † ◊	0.60 ± 0.12 †	-	0.54 ± 0.06	−6.2 ± 6.8	2.9 ± 5.6	-	-	-	-	5.6 ± 19.8 ◊	-	20 ± 11 ^†^ ◊		
	Cimolin et al., 2011 [33]	59	*n* = 12	Preoperative (*n* = 12)	0.54 ± 0.26	0.33 ± 0.12	0.17 ± 0.04	-	-	7.44 ± 2.24	7.92 ± 1.98	14.36 ± 4.52	43.18 ± 9.48	13.55 ± 8.66	15.83 ± 7.43	9.03 ± 3.24	-		
			*n* = 12	Short Term (10 m post-op)	0.84 ± 0.29 *	0.35 ± 0.11	0.13 ± 0.04 *	-	-	8.74 ± 2.96	8.68 ± 2.17	15.72 ± 7.77	40.66 ± 6.96	7.07 ± 7.93 *	5.02 ± 6.72 *	7.86 ± 3.26 *	-		
**Surgical Intervention**				**Kinematic Variable**									**Kinetic Variable**			**Bone Measurement**			
				**Knee**		**Ankle**				**Foot**									
**Surgical Procedure**	**Study**	**Q-Score** **/100**	** *n* **	**Group**	**K.Flex.Ext.IC**	**K.Min.Flex.St**	**K.Max.Flex.Sw**	**K.Amp.Flex.Ext**	**Ankle.Flex.Ext.IC**	**Ankle.Flex.St**	**Ankle.Min.St**	**Ankle.Flex.Ext**	**Foot.Mean.Progr.Adj**	**Hip.Ext.Max.Mm**	**Hip.Mean.Abd.Mm**		**Values**	**Assessment Method**	**Bone Variable**
Femoral Derotation Osteotomy	Boyer et al., 2017 [34]	60	*n* = 140	Preoperative	-	-	-	-	-	-	-	-	-		0.031 ± 0.029		50 ± 15	PE (Ruwe)	FT
			*n* = 140	Short Term (9–24 m post-op)	-	-	-	-	-	-	-	-	-		0.032 ± 0.031		15 ± 10 *		
			*n* = 29	Preoperative	-	-	-	-	-	-	-	-	-		0.036 ± 0.035		50 ± 15		
			*n* = 29	Short Term (9–24 m post-op)	-	-	-	-	-	-	-	-	-		0.038 ± 0.038		15 ± 10 *		
			*n* = 29	Mid-Term (>36 m post op)	-	-	-	-	-	-	-	-	-		0.040 ± 0.029 †		20 ± 11 ^†^ ◊		
	Cimolin et al., 2011 [33]	59	*n* = 12	Preoperative (*n* = 12)	27.46 ± 7.12	14.15 ± 5.62	48.78 ± 6.27	32.07 ± 7.27	3.75 ± 6.95	11.98 ± 5.42	−0.28 ± 5.49	12.26 ± 5.75	−0.81 ± 6.01	0.67 ± 0.19			-		
			*n* = 12	Short Term (10 m post-op)	24.87 ± 5.67	14.05 ± 6.70	50.56 ± 5.83	34.25 ± 6.93	2.95 ± 5.73	12.47 ± 6.33	−1.44 ± 8.09	13.91 ± 5.57	−9.39 ± 5.77 *	1.11 ± 0.17 *			-		

Non-significant difference: -; Significant difference between: preoperative and short term: *; preoperative and mid-term † ; short term and mid term : ◊.

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
