# Peer review of "Bone Deformities through the Prism of the International Classification of Functioning, Disability and Health in Ambulant Children with Cerebral Palsy: A Systematic Review"

_children, 2024, doi:10.3390/children11020257_

Round 1

Reviewer 1 Report

Comments and Suggestions for Authors

This is a systematic review reporting on the evidence regarding the relationship between lower limb bone morphology and body function, activity, and participation outcomes in ambulant children with cerebral palsy. The study also investigates the impact of changes in bone morphology, specifically induced by surgery, on these domains.

While the primary manifestations of CP are related to motor control and coordination, it is increasingly recognized that CP has far-reaching effects on multiple body systems, including the skeletal system. This review covers an important topic for clinicians and rehabilitation professionals who are caring for those children. However, I have several concerns that need to be addressed. Please, find them listed below.

1.     To contextualize the review and provide a foundation for understanding the significance and implications of the topic, authors may need to provide a comprehensive overview of the bone morphology changes observed in children with CP, highlighting their etiology, prevalence, and implications for clinical management.

2.     It is unclear which bone morphology changes this review focused on (i.e., alterations in bone mineral density, bone geometry, or bone quality).

3.     It is imperative to position the research question within the existing literature, elucidating any gaps, controversies, or unanswered questions in the field that the review seeks to address. This practice serves to establish the rationale and motivation for conducting the study.

4.     I suggest merging the third and fourth paragraphs in the introduction. Also, the fifth and sixth paragraphs.

5.     The clarification of eligibility criteria is essential to establish transparency and reproducibility. It is recommended that authors provide a more explicit demonstration of the measures taken to ensure the establishment of explicit and reproducible criteria for the selection of articles that have been included in the review. It would be of particular interest for authors to provide a detailed definition of the inclusion criteria, thus offering a comprehensive understanding of the specific parameters employed to determine the eligibility of articles.

6.     The systematic review incorporates a restricted number of studies, with only 12 studies meeting the pre-defined inclusion criteria out of a pool of 3,373 articles screened. This limited breadth of studies has the potential to impact the generalizability of the findings and constrain the depth of analysis. This should, therefore, be acknowledged in the discussion section.

7.     The authors mentioned that the meta-analysis was not possible and instead opted for a descriptive synthesis. This approach may constrain the capacity to draw definitive conclusions and obtain a comprehensive overview of the available evidence. It is also important to recognize this limitation, as it may impact the robustness and breadth of the findings presented in the study.

8.     The systematic review assessed the quality of the included studies using the Checklist for Case Series (CCS) and a quality assessment developed by Quebec University Hospital. However, the report does not provide detailed information on the quality assessment results, making it difficult to evaluate the robustness of the included studies.

9.     And more, the heterogeneity of study designs among the included studies. Variations in study methodologies and outcome measures may affect the comparability and generalizability of the findings.

10.  Could authors provide a more detailed interpretation of the correlations between bone morphology and body functions, discussing their clinical significance and implications for functional outcomes?

11.  While some limitations of this review have been mentioned, the discussion section could provide a more comprehensive reflection on these limitations. Additionally, it would be valuable to suggest specific areas for future research, such as the need for large-scale, well-designed studies to investigate the long-term effects of bone morphology and surgical interventions on functional outcomes, as well as the integration of qualitative research to explore the psychosocial aspects of bone deformities in children with cerebral palsy.

Reviewer 2 Report

Comments and Suggestions for Authors

The paper presented for the review is aimed to report the evidence for relationships between lower limb bone morphology and body functions, activity, and participation in ambulant children with CP, and to report the effect of changes in bone morphology (induced by surgery) on these domains.

The aim itself sounds like the Holly Grail of the whole neuroortopedics. And it would be the real miracle if the answer would be clear and simple…

Even the basic knowledge of the literature and empirical evidence suggest that the Holly Grail is empty with variations from no data to no or minor effect. 

First, the fact that “No studies evaluated the relationship between a lower limb bone morphology variable and an activity or participation outcome” makes doubtful the utility of ICF as the reference for the study protocol. In terms of body function, the authors meticulously analyzed the 3DGA data although these data are more close to the anatomy than the function. Among the generally accepted functional measurements the instruments like GMFM and FMS are more relevant. The authors didn’t specify GMFCS levels of the patients within the groups. It is important because function and activity are different between ambulating children who have levels 1, 2 and 3.

The authors used the term “bone morphology” to describe the bone structure. It is formally possible to attribute the term “morphology” to macro-anatomy but in the real world it’s more relevant to the micro-anatomy and bone architecture. Even agreeing to review I was intrigued by the title of the paper and awaiting for the histological data. 

Among the anatomical characteristics (which the authors call morphology) 3 major parameters (tibial and femoral torsion + NSA) are actually assessed. 

And when discussing femoral torsion the authors point out the question “What stops this child from playing, when she or he really wants to?” it seems quite strange to read it because the variables analyzed are not related to this question.

The general design of the paper seems questionable. Nevertheless the authors conducted an enormous work which is respectful. But the current title and the current aim of the paper seem misleading. It is recommended to rearrange the paper according to the existing data correlation of FT, TT and NSA with gait and selected functional parameters and place the ICF speculations in the discussion.

Round 2

Reviewer 1 Report

Comments and Suggestions for Authors

The authors have diligently addressed all the raised concerns and incorporated the suggested changes in a clear and comprehensive manner, resulting in substantial improvements to the manuscript. Based on the revisions made, it appears that the paper is now poised for publication and will significantly contribute to the field.

Reviewer 2 Report

Comments and Suggestions for Authors The authors followed the reviewer’s comments and modified the paper according the recommendations. The paper is recommended for publication in the present version.